

# Energy optimization for wireless sensor network using minimum redundancy maximum relevance feature selection and classification techniques

Muteeah Aljawarneh[1,2,*], Rim Hamdaoui[1,2,*], Ahmed Zouinkhi[2], Someah Alangari[1] and Mohamed Naceur Abdelkrim[2]

[1] Computer Science Department, College of Science and Humanities, Dawadmi, Shaqra University, Dawadmi, Riyadh, Saudi Arabia
[2] MACS Laboratory: Modeling, Analysis and Control of Systems, National Engineering School of Gabes, University of Gabes, Gabes, Tunisia
[*] These authors contributed equally to this work.

## ABSTRACT

In wireless sensor networks (WSN), conserving energy is usually a basic issue, and several approaches are applied to optimize energy consumption. In this article, we adopt feature selection approaches by using minimum redundancy maximum relevance (MRMR) as a feature selection technique to minimize the number of sensors thereby conserving energy. MRMR ranks the sensors according to their significance. The selected features are then classified by different types of classifiers; SVM with linear kernel classifier, naïve Bayes classifier, and k-nearest neighbors classifier (KNN) to compare accuracy values. The simulation results illustrated an improvement in the lifetime extension factor of sensors and showed that the KNN classifier gives better results than the naïve Bayes and SVM classifier.

## INTRODUCTION

The wireless sensor network (WSN) consists of a main sink and several sensors that collect data involving sound, temperature, pressure, humidity, and location. Which have been used in different areas like the military, health care, monitoring systems, *etc* (*Park et al., 2013*). WSN energy management is a hot spot caused by large deployed sensors over a wide area and limited battery capacity. In a sensor network, the lifetime of the sensor is very limited due to the very restricted power source. As a result, conserving energy and keeping it at a minimum level is usually a basic issue (*Alwadi & Chetty, 2012*). There are several approaches used to optimize energy in WSN, classified into five classes, namely: data reduction, routing, duty cycling, medium access control (MAC), and machine learning (ML) (*Aljawarneh et al., 2022*).

Corresponding authors
Muteeah Aljawarneh,
muteeah@su.edu.sa
Rim Hamdaoui,
r.hamdaoui@su.edu.sa

ML is very important in WSNs because it is used to upgrade performance and limit reprogramming and human intervention. Furthermore, it can access a large amount of data aggregated by the sensors. Besides that, the process of selecting useful information from collected data is not an easy process without ML. In addition; it is used in cyber-physical systems (CPS), integrating the Internet of Things (IoT), and machine to machine (*Kumar, Amgoth & Annavarapu, 2019*). ML is a mechanism that uses its prior experience to learn. It is implemented at the sink to improve the service quality at WSN (*Pundir et al., 2021*). ML techniques have been classified into four types: unsupervised learning, supervised learning, semi-supervised learning, and reinforcement learning. In our work, we will focus on supervised learning approaches that are classified into regression and classification. Classification is divided into perceptron-based (deep learning and ANN), logic-based (random forest and decision tree), instance-based (kNN) algorithms, statistical learning, SVM and Bayesian (*Kumar, Amgoth & Annavarapu, 2019*).

In machine learning development, a piece of rich information is obtained from the research topic, but a problem of irrelevant and redundant features appears, which causes higher feature dimensions. In classification learning, choosing the best learning samples is the primary key to training the classifier, where the redundant and irrelevant data will maximize classification algorithm complexity. Feature selection (FS) is a technique that selects the most favorable feature subset from the raw feature set to minimize dimensional space features. Its goals are to simplify the structure of data and upgrade the robustness and stability of the model (*Xie et al., 2023*).

The contribution of this work is to choose one of the best feature selection methods by comparing two methods, namely MRMR and Independent significant feature test IndFeat, an algorithm that ranks features according to their importance of use and removes the less significant input features on the output class; MRMR is a feature selection technique used to select features that have a low correlation between them and a high correlation with the output (class) and then reduce the input features to the classifier, which reduces the required time for processing, which increases the lifetime of a sensor node. After that, three classifiers are applied, which are the SVM with linear kernel classifier, the naïve Bayes classifier, and the k-nearest neighbor's classifier (KNN) to compare results. The performance of the classifier algorithms used is measured by using the confusion matrix.

The article is structured as follows: in the Literature Review section, a literature review of feature selection techniques and WSN energy issues is presented. The Materials and Methods section illustrates the proposed idea. Firstly, the MRMR algorithm is presented, and then the classification algorithm in our research is explained. Then the experimental results and a comparison between our work and previous work are illustrated in the Results and Discussion section. Finally, the conclusion and some perspectives are conducted in the Conclusion section.

# LITERATURE REVIEW

Various smart models have been proposed to optimize the energy efficiency in WSN; in *Barnawi & Keshta (2016)*, the authors present a study for three smart models depending on multilayer perceptron, support vector machine, and naïve Bayes classifiers. The results of the simulation showed that, for the same lifetime extension factor, the SVM classifier chooses sensors that give better energy efficiency if compared with the naïve Bayes and MLP classifiers. In *Alwadi & Chetty (2012)*, the authors introduced an algorithm that ranks the sensors from the most to least significant and then uses a naïve Bayes classifier, where, several datasets with various features are used to show the effect of feature ranking and selection on optimizing the energy. *Barnawi & Keshta (2014)* presented a smart model to optimize energy in WSN by using the classification algorithm multilayer perceptron (MLP) neural network, where the accuracy is improved and has a better result than the naïve Bayes classifier. The work presented in *Zouinkhi, Flah & Mihet-Popa (2021)* proposed an intelligent approach involving a cluster tree topology that aims to increase the lifespan of nodes connected to a wireless sensor network. A specifically different duty cycle is applied to each subgroup, depending on the level of energy remaining in the battery, which helps create a wide range of functional modes. The proposed approach proves the efficiency of the energy benefit. In *Kang et al. (2020)*, the authors concentrated on reducing the energy in the sensor itself in two stages. First, they applied hybrid filter-wrapper feature selection to find the minimum number of sensors that were chosen to guarantee the performance of WSN. Second, they focused on saving energy on each node by manipulating the sample rate and the interval of transmission. To achieve that, they proposed an optimization approach based on the Simulated Annealing (SA) algorithm. The proposed method finds the approximate global optimum in datasets where accurate values are difficult to collect due to noise issues, like sensor data. *Azzouz et al. (2022)* adopted algorithms based on unsupervised machine learning to improve the Leach routing protocol. They proposed an energy-aware cluster head selection protocol with balanced and fuzzy C-mean clustering in WSN. A new cluster head selection mode is adopted to solve the energy consumption problem in the LEACH protocol.

In *Rao, Jana & Banka (2017)*, the authors suggested an algorithm based on cluster head selection to optimize energy depending on partial swarm optimization (PSO), named PSO-ECHS. An efficacious scheme of fitness function and partial encoding is used to develop the algorithm. They tested the algorithm intensively in several scenarios by altering the number of wireless sensors and the CHs. Their work proved the efficiency of the suggested algorithm compared with other existing algorithms. *Ahmad et al. (2021)*, illustrated approaches for optimizing denial of service (DOS) anomaly detection and energy reservation in WSN to equilibrium them. They introduced a new clustering approach named the CH Rotations algorithm, to optimize the efficiency of anomaly detection over a WSN's lifetime. Also, they used a feature selection approach with a machine learning algorithm, to examine the traffic of the WSN node and evaluate the impact of these approaches on the lifetime of the WSN. *Jeevaraj (2023)* focused on finding a solution to the problem of attacks that occur from several types of threats that will

breakdown the network. The idea of the proposed system is to provide a solution to this type of attack and detect if there is abnormal behavior in any sensor. The processing of the data set is done by applying the minimum number of features for the intrusion detection system using ML algorithms. The main goal of this research is to optimize the prediction of intrusion at sensor nodes based on artificial intelligence algorithms. This also includes a feature selection method to increase the built model's performance by using the chosen classifier, as they use the Bayes category algorithm. The authors in *Pavone et al. (2023)* presented a generalized algorithm that depends on the principle of feature selection to accelerate the process of the artificial intelligence unit design. The work compared several feature selection approaches by computing accuracy, where the author tested the proposed approach on real water quality monitoring by using the XGBoost algorithm and a feature selection ranker approach by using the Weka simulator. The result showed that XGboost has the best performance. In *Raj & Duraipandian (2023)*, the authors proposed a unique unsupervised neural network method named Partly-Informed Sparse Auto Encoder (PISAE) that attempts to reset all sensors read from the selected prime numbers. In this study, two approaches of optimization methods were hybridized, which are the Harmony Search algorithm and Bacteria Foraging Optimization. Besides, they proposed a cross-layer based opportunistic routing protocol (CORP) as a routing protocol for WSN, used to choose the best path, reduce the computation time and the used energy, and improve the data transmission. *Yadav, Sreedevi & Gupta (2023)* worked to solve the issues of security and threats faced in WSNs. The author proposed a novel method that combined a feature selection approach called correlation-based feature selection (FCBFS) with XG-Boost. It is used to choose the best features in a cluster-based WSN before using the classifier. Five common machine learning-based classifiers are used, which are: random forest, decision tree, extra tree, naïve Bayes, and XGBoost, to develop a robust intrusion detection system in WSN and IoT applications. The effectiveness of the proposed approach is proven using accuracy, recall, precision, and the F-score. In *Parameshachari & Manjunath (2023)*, the authors used feature selection for IDS in mobile *ad hoc* networks (MANETs) applied to the NSL-KDD dataset as input data. They used two methods SMOTE and Z-score in pre-processing to strip the irrelevant features. The artificial butterfly algorithm is then used to carry out the best features. Finally, an adaptive voting mechanism is designed to choose the best classifier. The result proves, by estimated accuracy, the effectiveness of the proposed model. *Qaiyum et al. (2023)* suggested a creative technique to analyze energy depending on extracting and classifying the features of micro-grid photovoltaic data cells utilizing a deep learning algorithm. The optimization of energy for the micro-grid is carried out by using a system of photovoltaic energy. The analysis of the data is carried out by feature classification using a Markov encoder model with Gaussian radial Boltzmann. The proposed system achieved a power analysis of 88% and an accuracy of 93%. *Mojtahedi et al. (2022)* constructed a feature selection technique that depends on a genetic algorithm (GA) and a whale optimization algorithm (WOA). They proposed a sample-based classification technique. A KNN classifier was used; the system was evaluated in terms of accuracy and showed good results compared with previous methods. In the approach used, the genetic algorithm and the Whale optimization algorithm correctly chose the related features for the

class label. The KNN classifier discovered the misbehaving node in the intrusion detection dataset in WSNs. *Kadian, Rohilla & Kumari (2023)* showed that IoT-WSN-dependent energy efficiency protocols have an important effort in improving the lifetime of WSN. The combination of machine learning approaches and routing protocolsis very efficient in achieving this point.

## MATERIALS & METHODS

### The proposed system

In this study, we propose a smart model to optimize energy consumption in WSNs. In the first stage; we apply the MRMR algorithm as a classification algorithm that is used to select the sensors from the most to the least significant. Here, we assume that each feature represents a sensor in WSN; reducing the number of sensors is equivalent to reducing the number of features. Besides, reducing the number of features/sensor minimizes the energy consumed by sensors. The purpose of this process is to quickly distinguish and dismiss weak features. So, the number of inputs to the selection process will be minimized and reducing the required time for the final selection procedure. We assume that a lifetime extension factor is expressed as shown in Eq. (1) (*Alwadi & Chetty, 2012*):

$$LT = \frac{overall \ number \ of \ sensor}{number \ of \ utilized \ sensor}. \tag{1}$$

Equation (1) demonstrates the relationship between the number of used features and the lifetime extension factor. If we reduce the number of used features, the lifetime extension factor will be increased. Therefore, using a suitable algorithm enables the selection of the most important features/sensors and eliminates redundant features while maintaining the same accuracy.

Feature selection is an effective technique for processing high-dimensional data and improving learning efficiency (*Cai et al., 2018*). FS detects the relevant features and dismisses the redundant and irrelevant elements to obtain a subset of features that accurately depict the main problem without affecting performance (*Ramírez-Gallego et al., 2017*). It is categorized across different standards: one of these criteria is based on the training dataset used (unlabeled, labeled, or partially labeled). Accordingly, the feature selection approaches are categorized into unsupervised, supervised, and semi-supervised feature selection, respectively. A good FS approach is evaluated according to its high accuracy and should have less computational overhead (time and space complexity) (*Cai et al., 2018*). In our research, we focus on the supervised feature selection algorithm MRMR.

### *MRMR for feature selection*

The data size such as video, text, and image, has grown exponentially because of the quick development of artificial intelligence and neural networks. Moreover, due to the complexity of the data volume and its multi-dimensionality, a problem of irrelevant data and redundancy appeared. As a result, extracting useful features from the data is a hot spot. Feature selection is an essential approach that is used to reduce the dimensionality of the data. Briefly, feature selection is used to minimize the dimension of features while increasing the training speed of the model (*Ren, Ren & Wu, 2022*).

As we know, in any original set, the features are divided into four groups, which are: (a) weak relation and redundant features; (b) completely noisy and irrelevant features, (c) very strong relevant features, and (d) features that have weak relation and are non-redundant. Supervised FS consists of two groups: (c) and (d). Where the redundancy in features and relevance can be converted to two optimization problems: minimum redundancy and maximum relevance, a classic standard for feature selection depending on redundancy and relevance analysis MRMR (*Cai et al., 2018*). MRMR is a feature selection technique used to select features that have a low correlation between them and a high correlation with the output/class (*Radovic et al., 2017*). The MRMR approach is considered a grateful filter across the field of ML as evidenced by large citations (*Ren, Ren & Wu, 2022*). Initially, this approach was used to classify the DNA microarray data, which is considered a challenge in the ML felid because of the very large number of features and the low number of samples. MRMR approaches are widely used in several fields, such as the analysis of the movement of eyes and the analysis of satellite images that have multispectral and gender classification. Furthermore, MRMR is used in the preprocessing step of various high-dimensional problems, such as image and text analysis (*Ramírez-Gallego et al., 2017*).

The MRMR algorithm depends on the principle that it chooses a group of features (low-redundant) that have the highest correlation with the output result (max-relevance) and the least correlation between features in the main feature set.

The maximum correlation feature is represented by Eq. (2) (*Ren, Ren & Wu, 2022*):

$$maxD(S,c), D = \frac{1}{|S|} \sum_{x_i \in S} I(x_i; c). \tag{2}$$

$S$ represents the feature subset that consists of features $\{x_i\}$. $D(S,c)$ represents the average value of the mutual information between the classification target $c$, and $I(x_i; c)$ symbolizes the $i$th characteristics in $S$.

A large amount of redundancy is created between the chosen features by max-relevance, which results in feature dependence. When a redundant has occurred between two features, the redundancy is eliminated by using Min-Redundancy, as shown in Eq. (3):

$$minR(S), R = \frac{1}{|S|^2} \sum_{x_i, x_j \in S} I(x_i, x_j). \tag{3}$$

The MRMR algorithm in Eq. (4) is produced by combining Eqs. (2) and (3):

$$max\varphi(D,R), \varphi = D - R. \tag{4}$$

Practically, if we obtain the feature subset $S_{m-1}$, we use the remaining $X - S_{m-1}$ subset to choose the $m$ feature. If you want to maximize $\varphi(\cdot)$, you should improve the corresponding incremental algorithm as shown in Eq. (5) (*Ren, Ren & Wu, 2022*):

$$\max_{x_j \in X - S_{m-1}} [I(x_j; c) - \frac{1}{m-1} \sum_{x_i \in S_{m-1}} I(x_i; x_j)]. \tag{5}$$

Here $I(x_j; c)$ symbolizes the $j^{th}$ characteristics in $X - S_{m-1}$.

In the second stage, three classifiers are applied which are: the SVM with linear kernel classifier, the naïve Bayes classifier, and the k-nearest neighbors classifier (KNN), which are described in the next section. Classification is the most popular technique that is used in machine learning to predict the value of a categorical feature depending on the value of another attribute. In other words, classification is used to predict the new data class based on the training data. Where the classifier learns from the specific dataset and then classifies the newly observed data into several groups or classes (*Priyam et al., 2013*). The redundant or irrelevant features increase classification algorithm complexity, along with training and prediction time. As a result, it is very important to choose the best feature that affects the object and remove redundant and irrelevant features to increase the classifier's efficiency.

The proposed system is shown in Fig. 1. We have two datasets inside the dataset block, which are the ionosphere and gas sensor array drift datasets. MRMR is applied in the first stage to select features/sensors from most to least significant. The second stage runs the selected classifier among three ones: SVM with a linear kernel, naïve Bayes, and KNN classifier.

### Classification algorithm

- K-nearest neighbors algorithm

The primary idea of the KNN approach is to predict test data labels based on the majority rules, initially choosing the closest training samples $k$ to the test samples. After that, predict the test sample according to the major class of the $k$ closest training. In other words, KNN requires calculating the distance of the whole training samples for every test sample in the operation of choosing the $k$ closest neighbors (*Deng et al., 2016*).

- Naïve Bayes algorithm:

The naïve Bayes algorithm is considered strongly for several reasons, such as its very easy and simple understanding, high accuracy, and speed of classification. As a classifier, it depends on Bayes' theorem with a hypothesis of characteristic conditional independence (*Ning, Junwei & Feng, 2019*). It supposes that the occurrence of each feature is independent of the occurrence of other features. That is why it is called Naïve, and it is widely used in classification due to its reasonable performance (*Agarwal et al., 2018*).

- Support vector machine (SVM)

Support Vector Machine is a vastly utilized classification algorithm in machine learning. The main purpose of SVM is to separate the various classes in the training set by using a surface that increases the margin between them. SVM can be used to obtain powerful and precise classification results, even in the case of input data that cannot be linearly separated. It uses kernels to map the data into higher dimensional space. Firstly, SVM was suggested for binary classification, and then it is used to solve a multi-category problem because it independently learns from the dimension of the attribute space and gets a unique result. Generally, there are two types of SVM, soft margin and hard margin. Soft margin is used to classify nonlinear datasets and hard margin is used for linear data (*Yalsavar et al., 2021*).

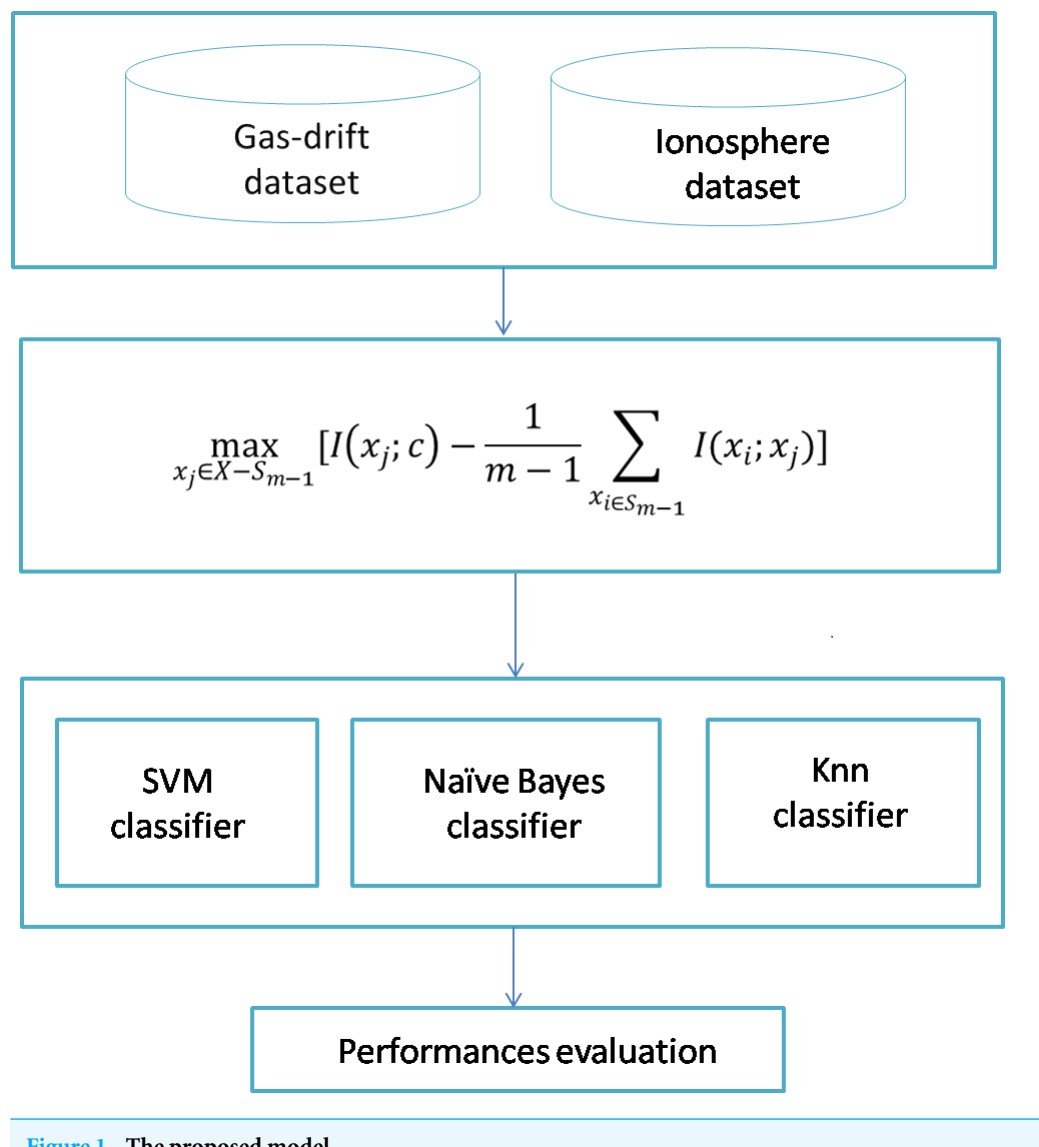

**Figure 1** The proposed model.

## RESULTS AND DISCUSSION

In our simulation, we studied two different datasets: the ionosphere and the gas sensor array drift. The datasets correspond to WSN in the UCI machine learning repository and have been used in many works (*Alwadi & Chetty, 2012*; *Alwadi & Chetty, 2015*). MATLAB (*The MathWorks Inc., 2022*) is used as a simulation tool. The datasets are summarized in Table 1.

The ionosphere dataset consists of 351 instances and 34 attributes; the source of the ionosphere dataset is a Labrador in a system goose bay that collects this data. It is collected by using 34 different real sensor nodes. This system is composed of a phased array of antennas

**Table 1  The used datasets.**

| Dataset | # number of instances | # number of attributes | Missing values | Associated task |
|---|---|---|---|---|
| Ionosphere | 351 | 34 | No | Classification |
| Gas sensor array drift | 13910 | 128 | No | Classification |

operating at a high frequency. The target of this dataset is free electrons in the ionosphere layer. The output of the dataset has two classes; good and bad. "Good" radar output means that this data presents proof of some kind of structure in the ionosphere. "Bad" output means that the data does not allow their signals to pass through the ionosphere (*Alwadi & Chetty, 2012*). The gas sensor array drift dataset consists of 13,910 measurements by 16 chemical sensors. The process can be presented as a classification task to discover the type of gas provided. In our dataset, we have 6 classes that represent the type of gas. The classes are ethanol, ammonia, ethylene, acetaldehyde, toluene and acetone (*Alwadi & Chetty, 2015*). The dataset has 128 features/sensors, and the class (or output) here is the type of gas.

- Experiment 1

This experiment was performed by applying the MRMR algorithm using an ionosphere dataset to select sensors from most to least significant, then using SVM with a linear kernel classifier. Simulation results show the accuracy using the confusion matrix when we select 10, 20, and 30 features, as shown in Figs. 2, 3, and 4, respectively. A confusion matrix is a two-dimensional matrix that is used to evaluate a system. This matrix evaluates the classifier's result when we apply the dataset by estimating the accuracy value. The accuracy is given by Eq. (6) (*Barnawi & Keshta, 2016*):

$$Accuracy = \frac{(TN + TP)}{(TN + TP + FP + FN)}. \tag{6}$$

The confusion matrix has four outcomes, which are true positive ($TP$), false positive ($FP$), true negative ($TN$), and false negative ($FN$).

- $TP$ occurs when the tested sample of the class is actually positive and is correctly classified as positive.
- $FP$ occurs when the tested sample of a class is actually negative and is incorrectly classified as positive.
- $TN$ occurs when the tested sample of a class is actually negative and is correctly classified as negative.
- $FN$ occurs when the tested sample of a class is actually positive and is incorrectly classified as negative.

The simulation results showed that the accuracy values are 90%, 94.3%, and 94.3% for 10, 20 and 30 features, respectively. The accuracy is calculated using the confusion matrix. By taking the sum of samples that are classified correctly, which are found in the diagonal of the matrix, and dividing the result by the overall number of instances in the examined dataset, we can see that the accuracies for the second and third selections are similar and

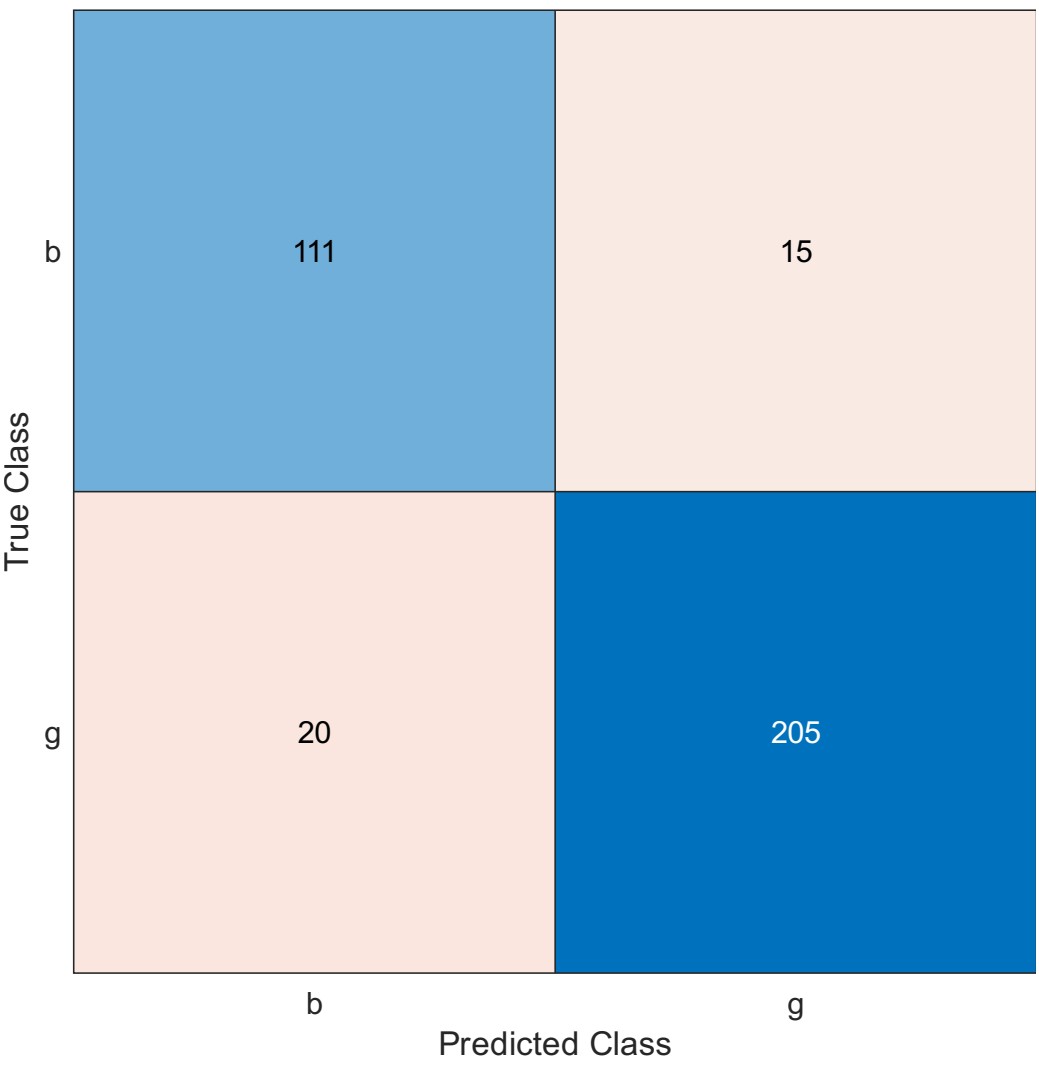

**Figure 2** **Confusion matrix of the kernel SVM when using the ionosphere dataset and selecting 10 features.**

better than the selection of 10 features. The lifetime extension factor when using 20 features is equal to 1.7; however, it is less when using 30, and it is equal to 1.13. The maximization of the lifetime extension factor means a reduction in WSN energy consumption.

• **Experiment 2**

In experiment 2, we used the ionosphere dataset, and the MRMR algorithm was applied to select the sensors from the most important to the least important then a naïve Bayes classifier was applied. The accuracy value is evaluated for 10, 20, and 30 features, as shown in Figs. 5, 6, and 7, respectively.

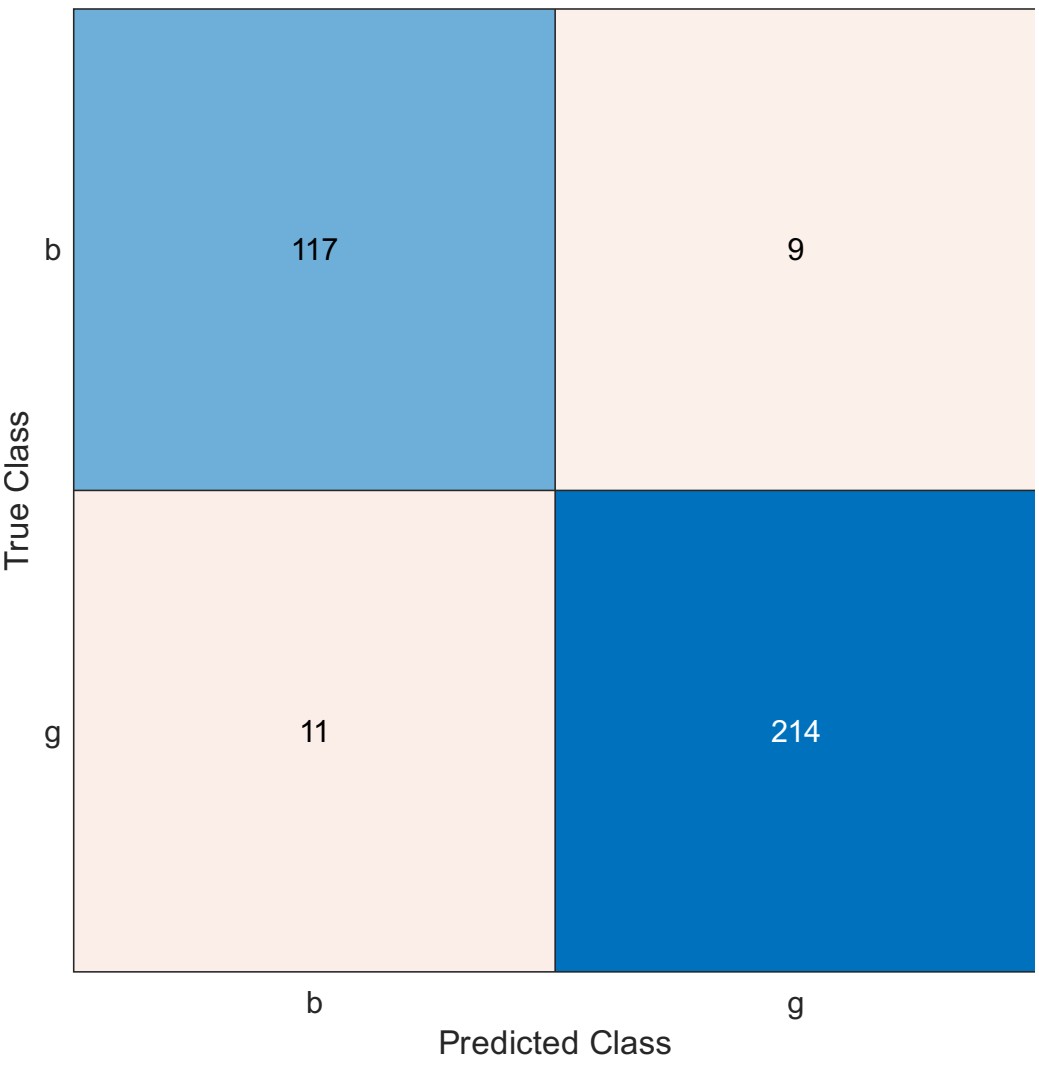

**Figure 3** **Confusion matrix of the kernel SVM when using the ionosphere dataset and selecting 20 features.**

Regarding kernel naïve Bayes, the accuracy results are 88.9%, 90.3%, and 91.5% for 10, 20, and 30. The results for 20 and 30 features are almost the same, which means it is possible to minimize the features/sensors without affecting the accuracy. However, if we compare it with the SVM classifier, SVM gives a better result.

- **Experiment 3**

This experiment was conducted on a gas sensor array drift dataset. In our experiment, we used the MRMR algorithm to select the features/sensors from the most to least significant, and after that, we applied it to the SVM and KNN classifier respectively. In this experiment, using the confusion matrix to evaluate the system for 128, 100, 90, 80, 70, and 60, features

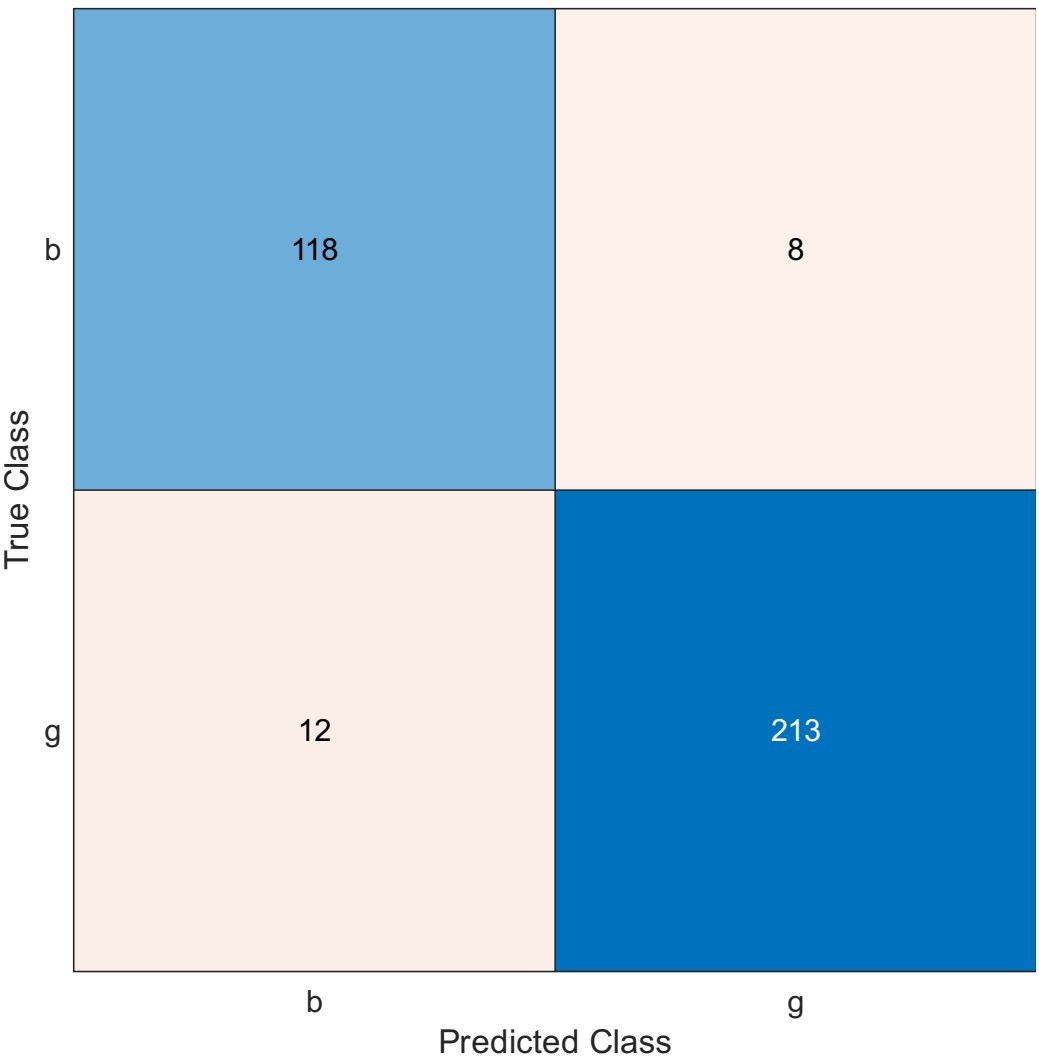

**Figure 4** Confusion matrix of the kernel SVM when using the ionosphere dataset and selecting 30 features.

depending on the estimated accuracy. The related confusion matrix for the SVM classifier is shown in the below figures from Figs. 8 to 14.

Tables 2 and 3 summarize the results of accuracy obtained for the SVM and KNN classifiers respectively. According to the SVM classifier results shown in Table 2, we can observe an increase in accuracy when we augment the number of features/sensors and obtain a reasonable value of accuracy when we reduce the number to 120, 100, and 90 features. The accuracy was reduced by only 0.2%. While for the KNN classifier results shown in Table 3, we see that we obtained superior results for the accuracy values. We have the same accuracy values for 128, 120, 100, 90, and 80 features/sensors which is equal to 99.5%. A reasonable accuracy value for 70, 60, and 40 features/sensors is given. We

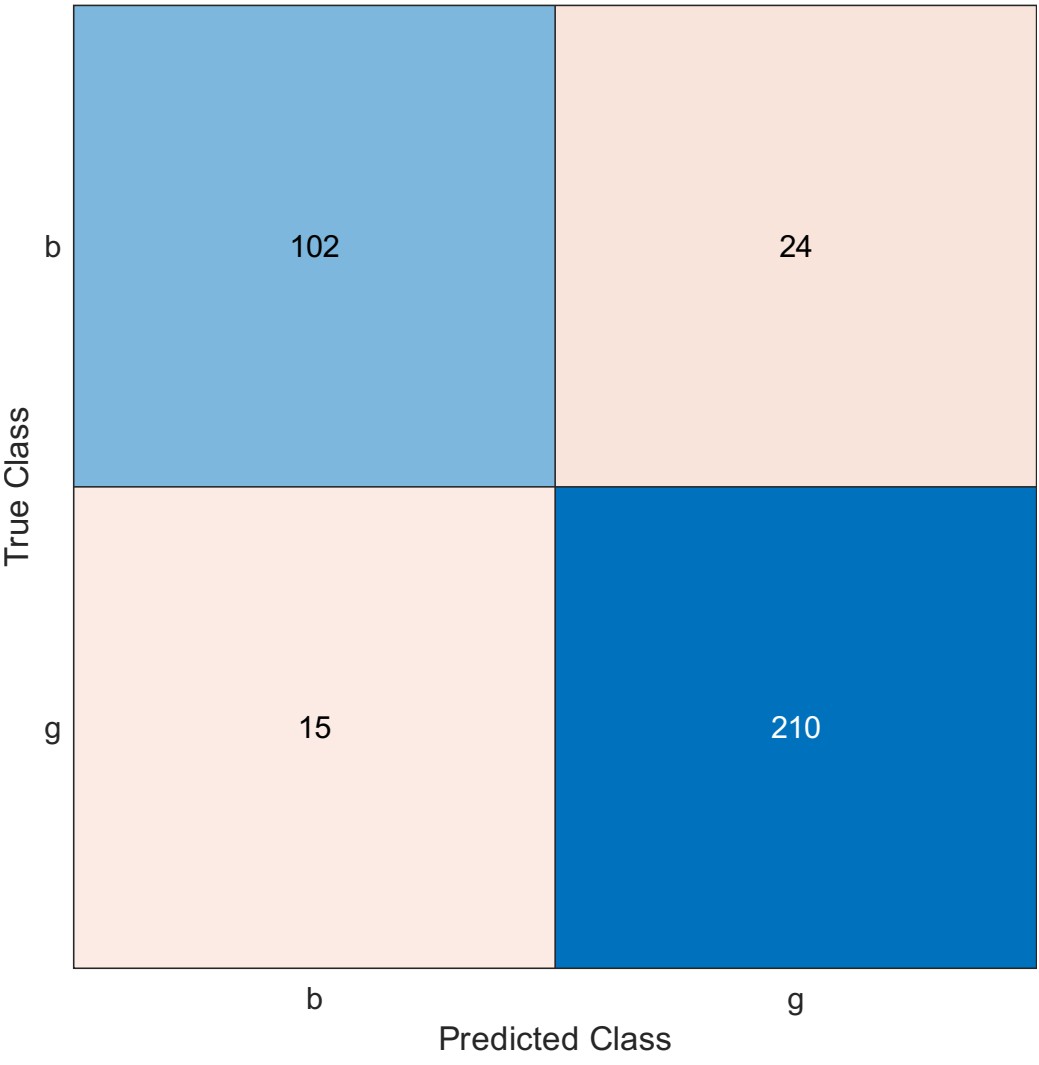

**Figure 5** Confusion matrix of the kernel naïve Bayes when using the ionosphere and selecting 10 features.

can conclude that the KNN classifier has better accuracy values than the SVM classifier. Then, by reducing the number of features/sensors, we reduce the energy consumed and the processing time, as well as increase the lifetime extension factor.

Table 4 shows a comparison between MRMR and IndFeat performed on the ionosphere dataset using naïve Bayes classification. We note from the simulation results that the MRMR gives better accuracy than the Endfeat. In Table 5, a comparison between MRMR and IndFeat using SVM linear classification has been conducted on the ionosphere dataset. We notice that the accuracy value is better for MRMR, with the accuracy value being 94.3%

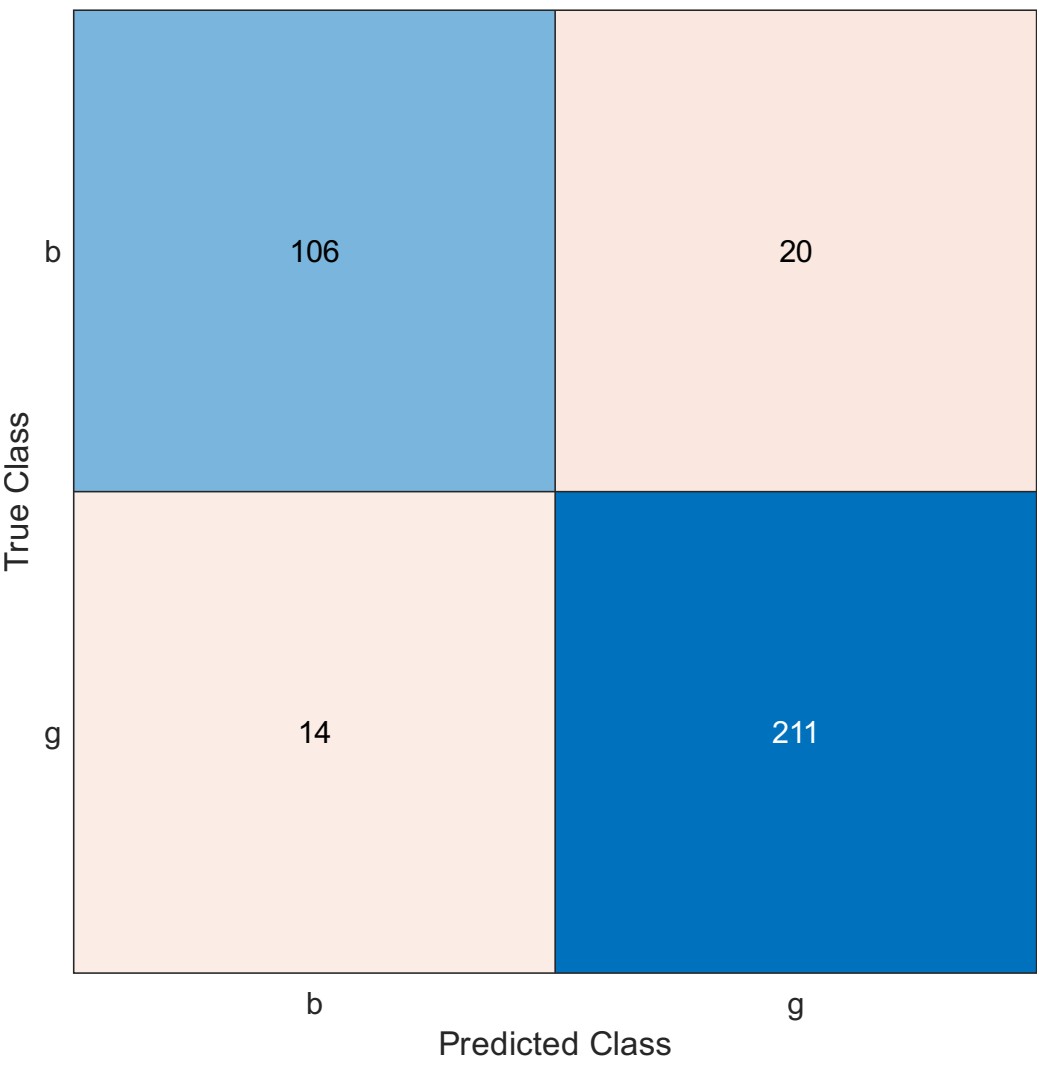

**Figure 6** Confusion matrix of the kernel naïve Bayes using the ionosphere dataset and selecting 20 features.

for 30 and 20 features/sensors. Therefore, we can reduce the number of features to 20 features/sensors and still obtain the same accuracy.

We conclude from the experimental results that the SVM classifier has a better result than the naïve Bayes classifier. Besides, accuracy results are better for the MRMR algorithm if it is compared to the IndFeat algorithm (*Xie et al., 2023*). Table 6 illustrates the lifetime extension factor LT and the accuracy values according to the number of used features/sensors. If we want to estimate it for experiment 3, when using the KNN classifier for 80 features out of 128, the LT is 1.6, and so on.

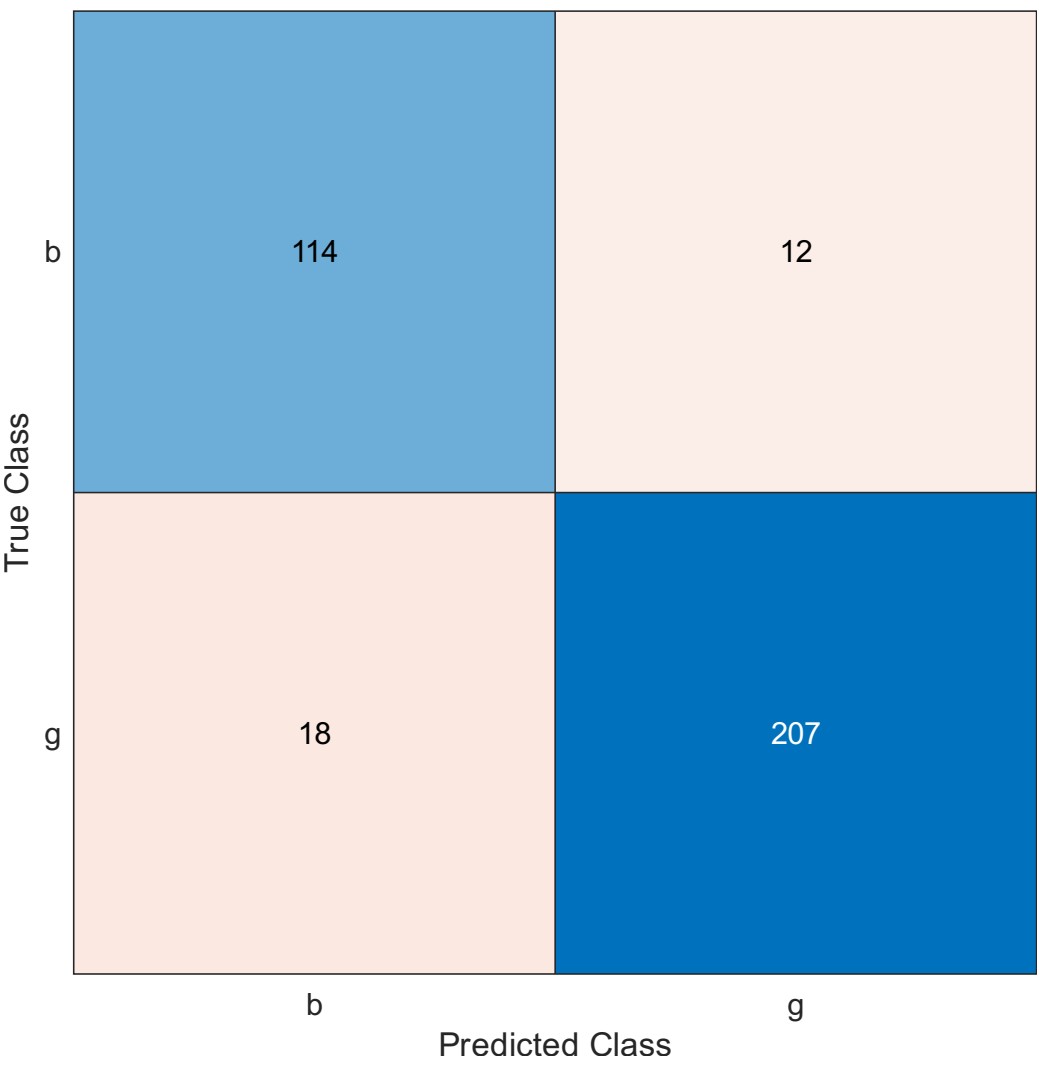

**Figure 7  Confusion matrix of the kernel naïve Bayes conducted on ionosphere and selecting 30 features.**

From Table 5, we can conclude that LT will increase as fewer features/sensors are used. However, there is a compromise between minimizing the features and conserving the accuracy that has to be respected.

Figure 15 shows the LT for experiments 2 and 3. We see the impact of reducing the number of features/sensors on increasing the lifetime extension factor. This result confirms the idea of this work and proves that MRMR is very effective in obtaining the maximum level of accuracy, increasing the lifetime extension factor of a sensor node, and improving energy consumption.

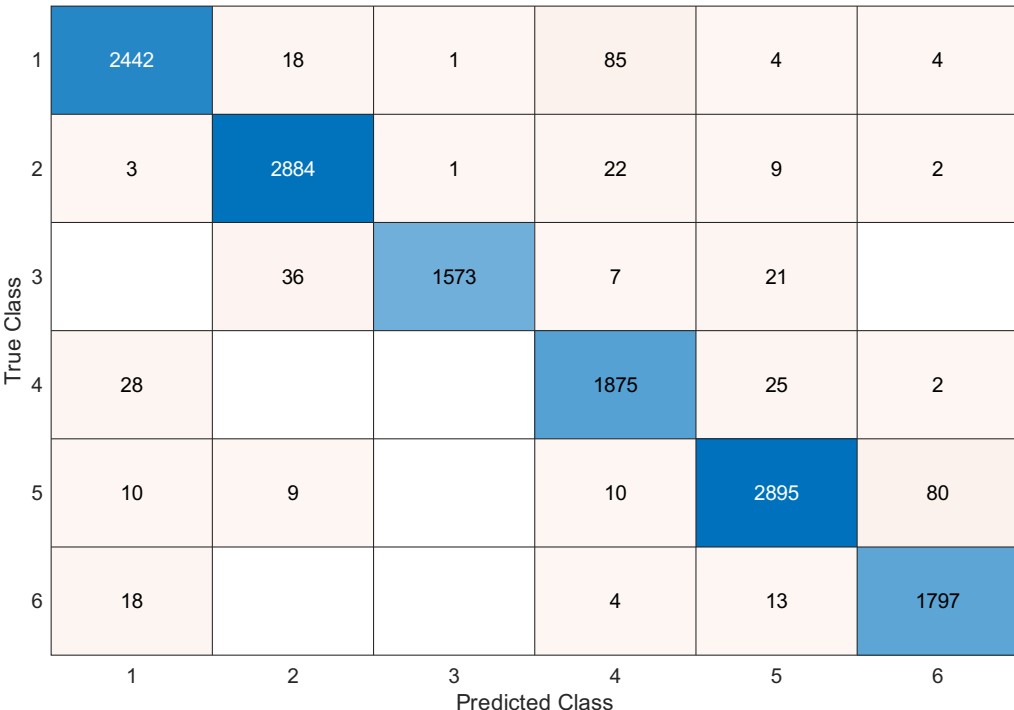

**Figure 8** **Confusion matrix of the SVM kernel conducted on gas sensor array drift dataset for 128 fea-tures.**

**Table 2** **Accuracy values for SVM classifier conducted on gas sensor array drift dataset.**

| Number of features | Accuracy |
| --- | --- |
| 128 | 97 |
| 120 | 96.9 |
| 100 | 96.8 |
| 90 | 96.8 |
| 80 | 96.7 |
| 70 | 96.6 |
| 60 | 95.6 |
| 40 | 91.1 |

**Table 3** **Accuracy value for KNN classifier conducted on gas sensor array drift dataset.**

| Number of feature | Accuracy |
| --- | --- |
| 128 | 99.5 |
| 120 | 99.5 |
| 100 | 99.5 |
| 90 | 99.5 |
| 80 | 99.5 |
| 70 | 99.4 |
| 60 | 99.2 |
| 40 | 99.1 |

**Table 4  Precision value comparison using naïve Bayes classification.**

| Number of features (N) | Accuracy using IndFeat | Accuracyusing MRMR algorithm |
|---|---|---|
| 10 | 76.02 | 88.9 |
| 20 | 80.8 | 90.3 |
| 30 | 81.4 | 91.5 |

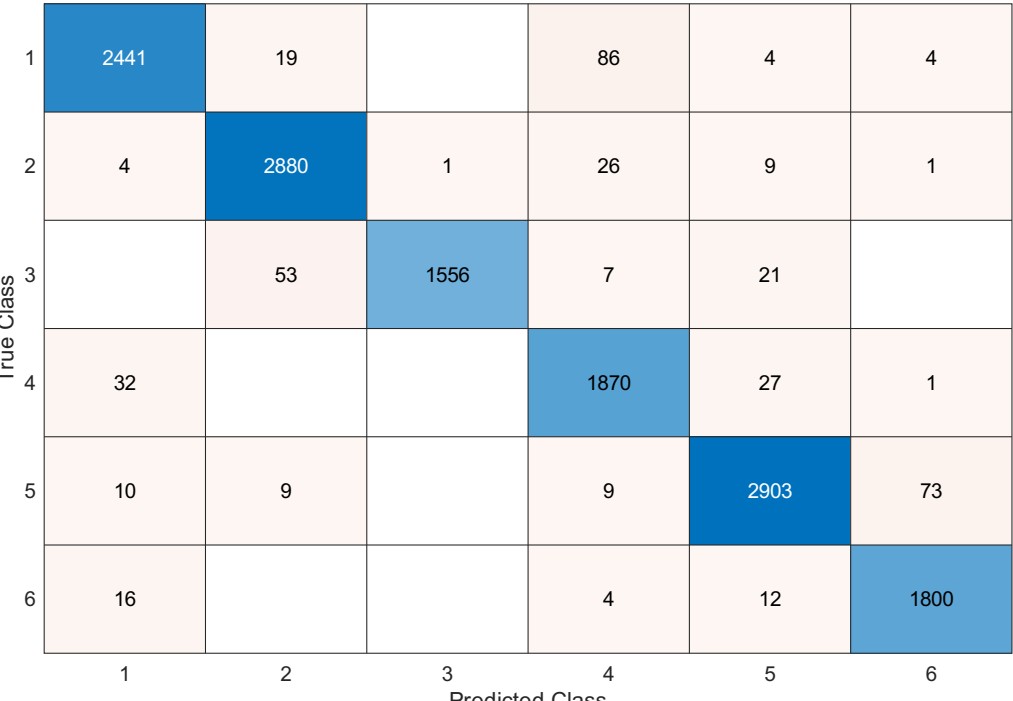

**Figure 9  Confusion matrix of the SVM kernel conducted on gas sensor array drift dataset for 120 feature.**

**Table 5  Precision values comparison using SVM linear classification.**

| Number of features ( N) | Accuracy using IndFeat | Accuracy using MRMR algorithm |
|---|---|---|
| 10 | 85.36 | 90 |
| 20 | 85.4 | 94.3 |
| 30 | 83.2 | 94.3 |

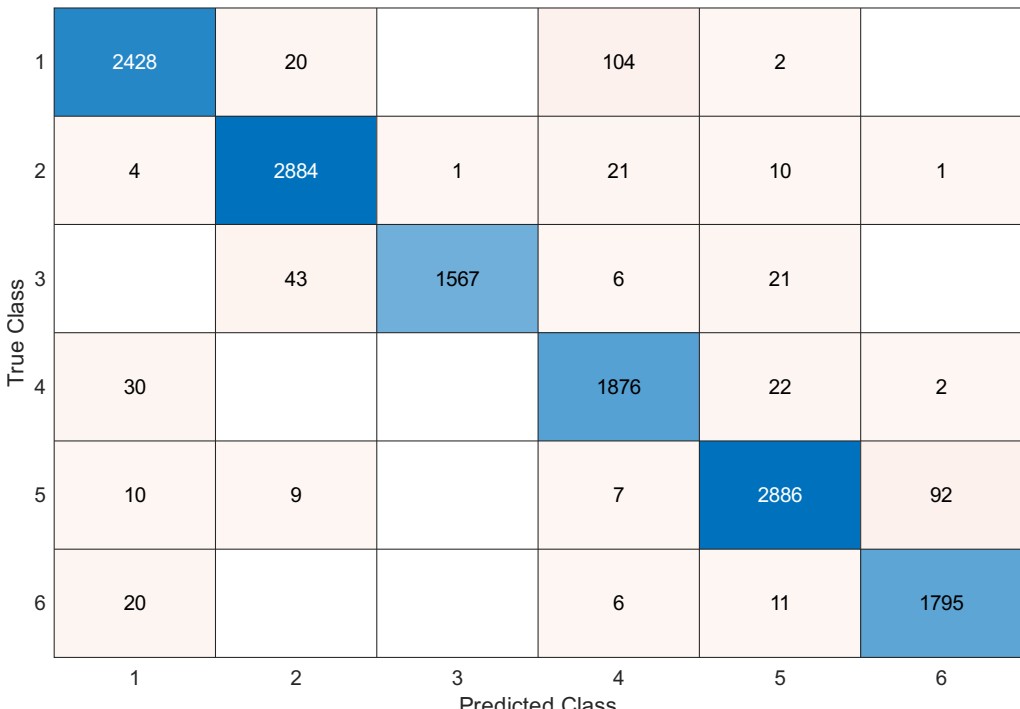

**Figure 10 Confusion matrix of the SVM kernel conducted on gas sensor array drift dataset for 100 features.**

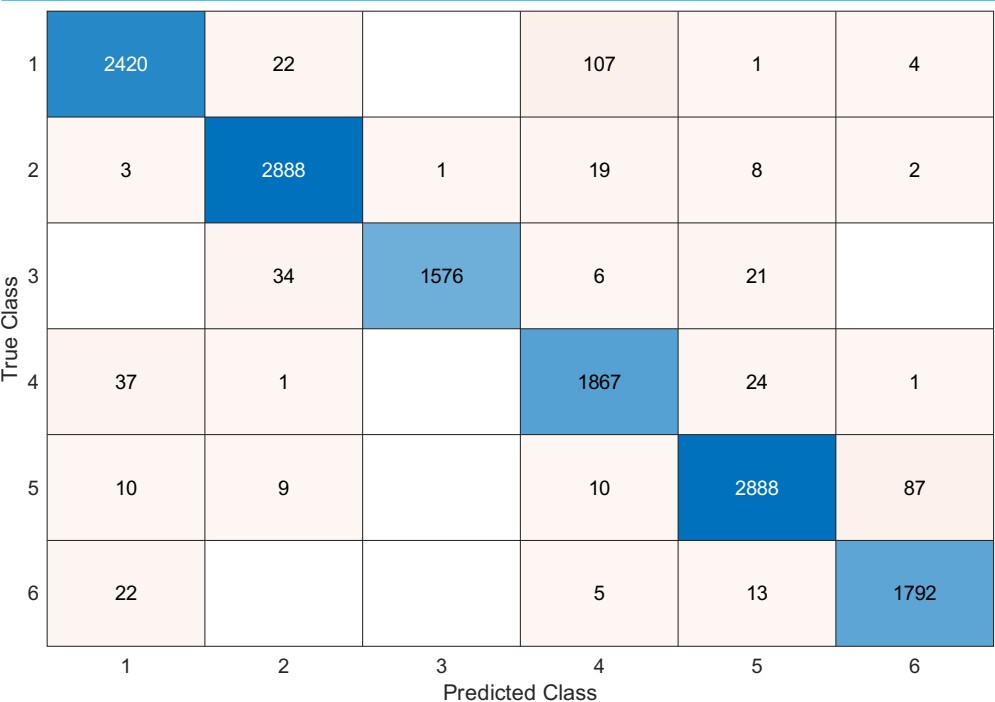

**Figure 11 Confusion matrix of the SVM kernel conducted on gas sensor array drift dataset for 90 features.**

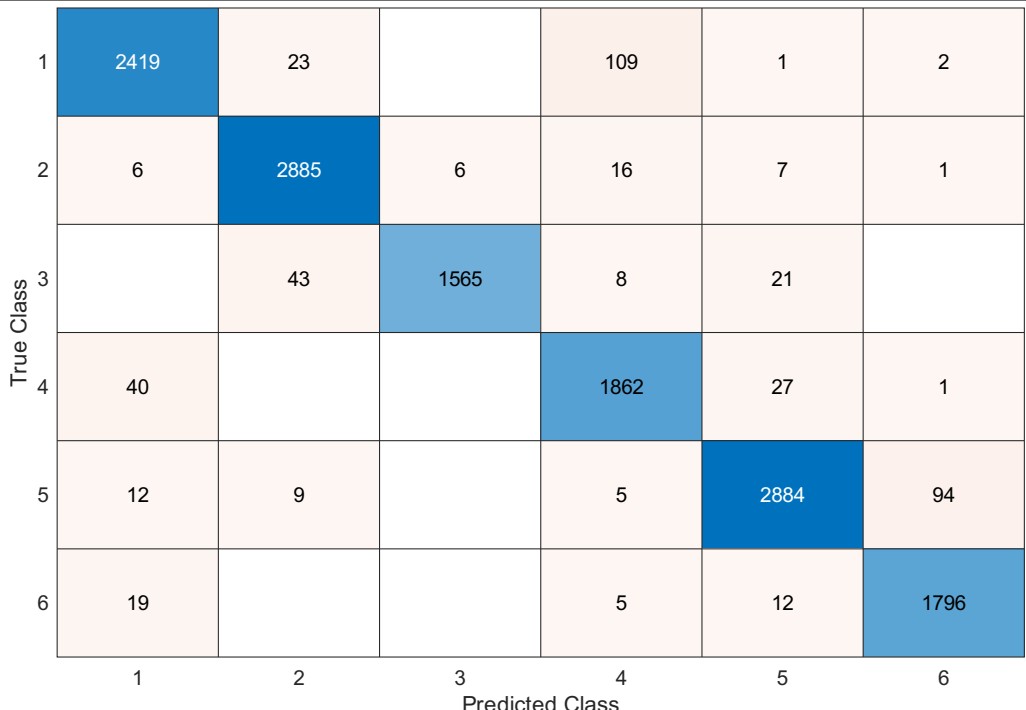

**Figure 12 Confusion matrix of the SVM kernel conducted on gas sensor array drift dataset for 70 features.**

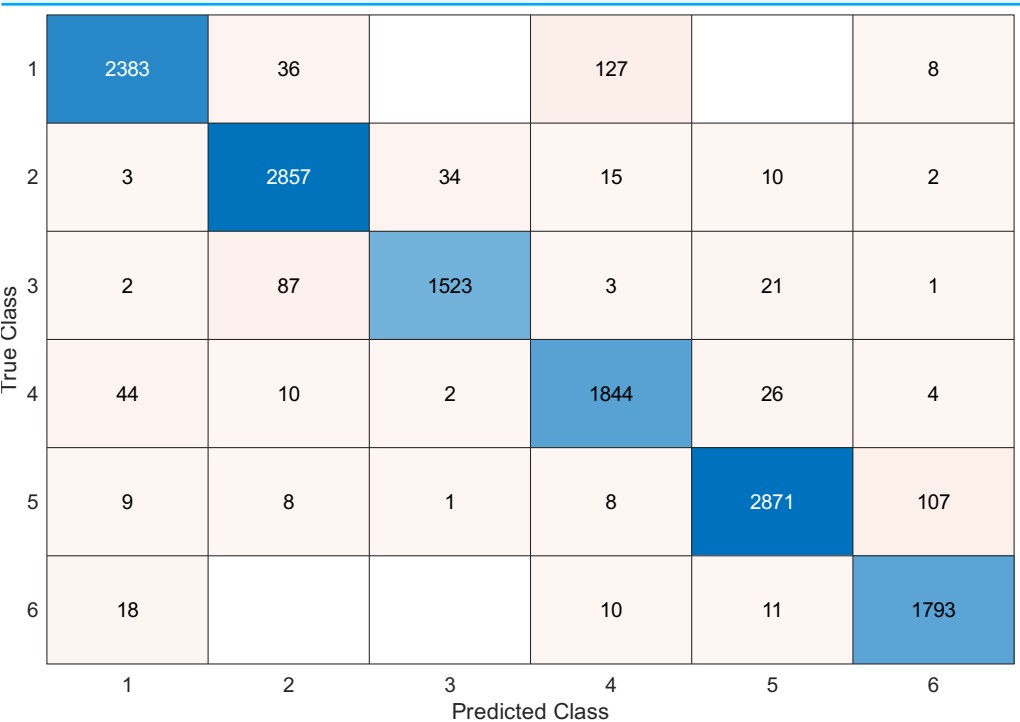

**Figure 13 Confusion matrix of the SVM kernel conducted on gas sensor array drift dataset for 60 features.**

| | 1 | 2 | 3 | 4 | 5 | 6 |
|---|---|---|---|---|---|---|
| **1** | 2289 | 87 | 57 | 114 | 2 | 5 |
| **2** | 7 | 2640 | 55 | 75 | 110 | 34 |
| **3** | 12 | 65 | 1514 | 20 | 24 | 2 |
| **4** | 130 | 45 | | 1699 | 42 | 14 |
| **5** | 19 | 43 | 4 | 28 | 2820 | 90 |
| **6** | 33 | 79 | | 29 | 17 | 1674 |

True Class — Predicted Class

**Figure 14** Confusion matrix of the SVM kernel conducted on gas sensor array drift dataset for 40 features.

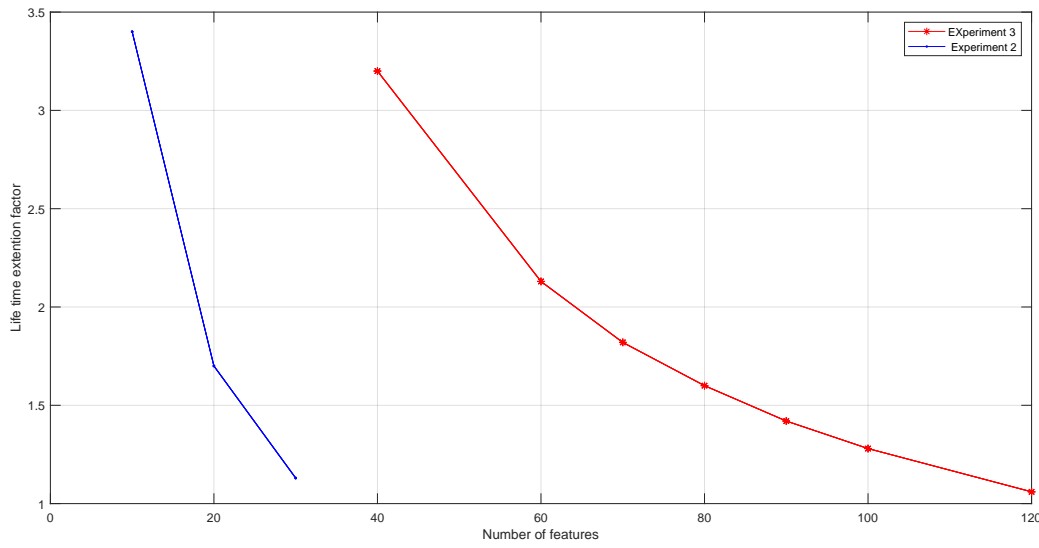

**Figure 15** Lifetime extension factor for Experiment 2 and Experiment 3.

**Table 6  The life time extension factor.**

| Number of used features | 120 | 100 | 90 | 80 | 70 | 60 | 40 |
|---|---|---|---|---|---|---|---|
| LT | 1.06 | 1.28 | 1.42 | 1.6 | 1.82 | 2.13 | 3.2 |
| Accuracy for knn classifier | 99.5 | 99.5 | 99.5 | 99.5 | 99.4 | 99.2 | 99.1 |

## CONCLUSION

This article presents an intelligent energy optimization model in WSN based on MRMR as a ranking algorithm to select the sensors from the most to least significant, followed by SVM, naïve Bayes, and KNN classifiers. We conclude that, by using the proposed feature selection approach, we can reduce the number of sensors without affecting the accuracy value. The lifetime extension factor is increased when we use fewer sensors. The simulation results showed that the KNN classifier gave better results than the naïve Bayes and SVM classifiers. The proposed approach, based on MRMR as a ranking method, showed better results when compared to other ranking techniques.

Future work will focus on reducing the complexity of the proposed system model and incorporating additional techniques to improve energy optimization in WSN.

### Funding

This APC was supported by the Deanship of Scientific Research at Shaqra University. The funders had no role in study design, data collection and analysis, decision to publish, or preparation of the manuscript.

### Grant Disclosures

The following grant information was disclosed by the authors:
The Deanship of Scientific Research at Shaqra University.

### Competing Interests

The authors declare there are no competing interests.

### Author Contributions

- Muteeah Aljawarneh conceived and designed the experiments, performed the experiments, analyzed the data, performed the computation work, prepared figures and/or tables, and approved the final draft.
- Rim Hamdaoui performed the experiments, analyzed the data, performed the computation work, prepared figures and/or tables, and approved the final draft.
- Ahmed Zouinkhi conceived and designed the experiments, analyzed the data, performed the computation work, authored or reviewed drafts of the article, and approved the final draft.
- Someah Alangari analyzed the data, authored or reviewed drafts of the article, and approved the final draft.

• Mohamed Naceur Abdelkrim analyzed the data, authored or reviewed drafts of the article, and approved the final draft.

## Data Availability

The raw measurements are available in the Supplementary Files.

## Supplemental Information

Supplemental information for this article can be found online at http://dx.doi.org/10.7717/peerj-cs.1997#supplemental-information.

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
