# Peer review of "Energy optimization for wireless sensor network using minimum redundancy maximum relevance feature selection and classification techniques"

_PeerJ Computer Science, doi:10.7717/peerj-cs.1997_

## Round 0.1 · original submission · Major Revisions

Remember to take into account the comments of the three reviewers.

IMPORTANT: This submission reports a machine learning study using free WEKA software (<https://www.cs.waikato.ac.nz/ml/weka/>). Please ensure that the workflow is described in enough detail to allow readers to reproduce the findings.

**Language Note:** PeerJ staff have identified that the English language needs to be improved. When you prepare your next revision, please either (i) have a colleague who is proficient in English and familiar with the subject matter review your manuscript, or (ii) contact a professional editing service to review your manuscript. PeerJ can provide language editing services - you can contact us at copyediting@peerj.com for pricing (be sure to provide your manuscript number and title). – PeerJ Staff

·

Basic reporting

The paper is good but can be improved further.

Experimental design

The experimental works can be improved, especially in terms of system complexity.

Validity of the findings

The authors must improve the paper quality by incorporating the suggested comments.

Additional comments

The manuscript have be reviewed and the following are some suggestions-
1. Ref. [27] to [30] must be cited in the manuscript.
2. The fig. (1) and fig. (15) picture quality can be improved further.
3. Most of the mathematical equations must be cited.
4. The authors must discuss all the mathematical equations.
5. The authors must discuss all tables.
6. The authors must discuss the system complexity.
7. The manuscript presentation can be improved further.

Reviewer 2 ·

Basic reporting

Work has not been prepared well

Experimental design

Work is not validated with standards

Validity of the findings

not validated

Additional comments

consistency missing

Reviewer 3 ·

Basic reporting

good in general but some formatting and alignment issues.

Experimental design

- You need to describe the tool (Weka) that you used for your experiments and adding reference for the tool you used.(I did not see the reference to WEKA).
- For Experiments 1, experiment 2 and experiment 3 sections in your paper, you should add table to describe each dataset you used in your experiments to clarify your intext writing.
- no reference seen or any intext writing to mention from where you got your Datasets for experiments!.

Validity of the findings

valid, datasets are very well known,

Additional comments

in case you are going to publish page 16 onwards , you need to format and bring up your figures without leaving any empty pages , see page 17 and 18 as an example you can combine those 2 in 1 page. The rest are the same (page 19 , page 20 , page 21 , page 22, page 23 , page 24 , page 25 , page 26 , and so on for the rest of figures.).

---

## Round 0.2 · accepted · Accept

Although the article is accepted for the final version.

·

Basic reporting

The Authors have incorporated the reviewer's comments in the revised manuscript.

Experimental design

no comment.

Validity of the findings

no comment.

Additional comments

The paper is good.
Acceptable in it's current form.
The Authors have incorporated the reviewer's comments in the revised manuscript.

Reviewer 3 ·

Basic reporting

feedback given was updated.

Experimental design

accepted but with minor revisions , all my comments have been amended in the paper.

Validity of the findings

valid